# Overexpression of the Ubiquitin Specific Proteases USP43, USP41, USP27x and USP6 in Osteosarcoma Cell Lines: Inhibition of Osteosarcoma Tumor Growth and Lung Metastasis Development by the USP Antagonist PR619

**DOI:** 10.3390/cells10092268

**Published:** 2021-08-31

**Authors:** Mélanie Lavaud, Mathilde Mullard, Robel Tesfaye, Jérôme Amiaud, Mélanie Legrand, Geoffroy Danieau, Régis Brion, Sarah Morice, Laura Regnier, Maryne Dupuy, Bénédicte Brounais-Le Royer, François Lamoureux, Benjamin Ory, Françoise Rédini, Franck Verrecchia

**Affiliations:** 1INSERM UMR1238 “Bone Sarcomas and Remodeling of Calcified Tissues”, Nantes University, 44035 Nantes, France; melanie.lavaud@etu.univ-nantes.fr (M.L.); mathilde.mullard@univ-nantes.fr (M.M.); robel.tesfaye@etu.univ-nantes.fr (R.T.); jerome.amiaud@univ-nantes.fr (J.A.); melanie.legrand2@gmail.com (M.L.); geoffroydn@gmail.com (G.D.); regis.brion@univ-nantes.fr (R.B.); sarah.morice@univ-nantes.fr (S.M.); laura.regnier@univ-nantes.fr (L.R.); maryne.dupuy@etu.univ-nantes.fr (M.D.); Benedicte.brounais@univ-nantes.fr (B.B.-L.R.); francois.lamoureux@univ-nantes.fr (F.L.); Benjamin.Ory@univ-nantes.fr (B.O.); francoise.redini@univ-nantes.fr (F.R.); 2CHU Hôtel Dieu, 44035 Nantes, France

**Keywords:** Ubiquitin Specific Proteases, Osteosarcoma, PR619

## Abstract

Osteosarcoma (OS) is the most common malignant bone tumor in children and teenagers. In many cases, such as poor response to treatment or the presence of metastases at diagnosis, the survival rate of patients remains very low. Although in the literature, more and more studies are emerging on the role of Ubiquitin-Specific Proteases (USPs) in the development of many cancers, few data exist regarding OS. In this context, RNA-sequencing analysis of OS cells and mesenchymal stem cells differentiated or not differentiated into osteoblasts reveals increased expression of four USPs in OS tumor cells: USP6, USP27x, USP41 and USP43. Tissue microarray analysis of patient biopsies demonstrates the nucleic and/or cytoplasmic expression of these four USPs at the protein level. Interestingly, Kaplan–Meyer analysis shows that the expression of two USPs, USP6 and USP41, is correlated with patient survival. In vivo experiments using a preclinical OS model, finally demonstrate that PR619, a USP inhibitor able to enhance protein ubiquitination in OS cell lines, reduces primary OS tumor growth and the development of lung metastases. In this context, in vitro experiments show that PR619 decreases the viability of OS cells, mainly by inducing a caspase3/7-dependent cell apoptosis. Overall, these results demonstrate the relevance of targeting USPs in OS.

## 1. Introduction

Osteosarcoma (OS) is the most common primary malignant bone tumor observed in children and adolescents [1]. The annual incidence of OS in the general population is five cases per million [2]. For several decades, the treatment of choice has been a combination of different drugs (methotrexate, doxorubicin, ifosfamide, cisplatin, etc.) before and after surgery [3,4,5]. Bad response to treatment and the presence of metastases at diagnosis remain the leading causes of death in patients with OS. An approximately five-year survival rate of 70–75% is observed for localized disease and patients responding well to chemotherapy, compared with only 20% for patients resistant to chemotherapy or with metastases at diagnosis [6,7,8]. Although some genetic factors, such as mutations or deletions in *P53*, have been identified, the exact etiology of this disease remains poorly understood [9,10]. Whole genome sequencing analysis show that OS is characterized by very high rates of genetic alterations with somatic mutations [11]. The lack of response to conventional treatments reported in many patients with OS highlights the urgency to identify new therapeutic strategies.

After synthesis, most proteins undergo modifications such as acetylation, methylation, phosphorylation, glycosylation, sumoylation or ubiquitination [12]. One of these modifications is ubiquitination, which allows a small molecule of ubiquitin to bind to lysine residues of target proteins [13]. All types of ubiquitination (mono-ubiquitination, poly-ubiquitination or branched ubiquitination) are crucial in driving mechanisms involved in cellular processes such as cell cycle regulation, response to DNA damage, chromatin remodeling and protein degradation by proteasome. Dysregulation of this process can therefore lead to the development of cancers [13,14]. Importantly, ubiquitination is a reversible process through the function of deubiquitinases (DUBs) that remove ubiquitin chains from target proteins, leading to the regulation of the stability or activity of these proteins [15]. To date, about 100 DUBs have been reported to be encoded by the human genome [16]. According to the organization of their catalytic domain, DUBs are classified into five classes: ubiquitin-specific proteases (USP), cysteine proteases, ubiquitin C-terminal hydrolases (UCH), ovarian tumor proteases (OTU), Machado–Joseph domain proteases (MJD) and JAB1/MNP/MOV34 metalloproteases (JAMM) [15]. USPs constitute the most numerous classes, with about 60 proteases in humans [17].

Primary tumor growth and metastatic progression may be associated with excessive protein synthesis or degradation. In this context, an increasing number of studies indicate that USPs regulate tumor formation by modulating key functions involved in cancer progression such as cell proliferation and death [14,18]. Literature is scarce concerning USPs and primary bone sarcoma, specifically OS. Silencing of USP1, previously reported to be overexpressed in some malignant tumors such as non-small cell lung cancer, inhibits the proliferation and invasion of U2OS OS cells [19]. USP7, overexpressed in many tumors such as hepatocellular carcinoma, regulates OS cells metastasis by inducing epithelial-mesenchymal transition [20]. Downregulation of USP22 suppresses OS tumor growth and metastasis in vivo [21]. Knock down of USP39 inhibits OS cell proliferation and induces apoptosis [22].

Although USPs seem to be particularly important in OS development, no large-scale transcriptomic studies, no correlation between USP expression and patient survival, and no evaluation of the effects of a USP inhibitor in a preclinical model have been performed to date.

In this context, the aims of this work were: (1) to compare the expression of USPs in OS cell lines and in mesenchymal stem cells differentiated or not into osteoblasts, (2) to explore USPs expression in patient biopsies, (3) to identify a potential correlation between the expression of some USPs and patient survival, and (4) to evaluate the effect of the USPs inhibitor PR619 on the growth of OS tumors using an orthotopic mouse model.

## 2. Materials and Methods

### 2.1. Cell Cultures

Human OS (HOS, KHOS, G292, MG63, U2OS, 143B, SAOS2) cell lines were cultured in Dulbecco’s Modified Eagle’s Medium (Lonza, Basel, Switzerland) with 10% fetal bovine serum (Hyclone Perbio, Bezons, France) and 1% penicillin–streptomycin (Lonza, Basel, Switzerland). Human mesenchymal stem cells (MSCs) were provided from the Institute for Clinical Transfusion Medicine and Immunogenetics (Ulm, Germany). This center has a production license for MSCs from bone marrow aspirates (production license DE-BW-01-MIA-2013-0040/DE-BW-01-IKT Ulm) using Good Manufacturing Practices according to defined standard operating procedures and in compliance with the established quality management system. MSCs were cultured in Minimum Essential Medium Alpha (1X) (Life technologies, Carlsbad, CA, USA) with 5% platelet lysate plasma, 1 IU/mL heparin sodium (Panpharma, Luitré, France) and 1% penicillin–streptomycin (Lonza). Mycoplasma level has been tested according to the manufacturer’s protocol (Lonza). PR619 was purchased from Sigma (St. Quentin Fallavier, France).

### 2.2. Real-Time Polymerase Chain Reaction

Total RNA cells were extracted using NucleoSpinPlus Kit (Macherey Nagel, Duren, Germany). Total RNA was used for first-strand cDNA synthesis using the Maxima H Minus First Strand cDNA Synthesis Kit (ThermoFischer, Illkirch, France). Real-time PCR was performed with a CFX 96 real-time PCR instrument (Biorad, Richmond, CA, USA) using SYBR Select Master Mix (ThermoFischer). Primers sequences are provided in Table 1.

### 2.3. RNA Sequencing and Analysis

Libraries were prepared at Active Motif Inc. using the Illumina TruSeq Stranded mRNA Sample Preparation kit, and sequencing was performed on the Illumina NextSeq 500 as 42-nt long-paired end reads (PE42). Fastp (v. 0.19.5, [23]) was used to filter low quality reads (Q > 30) and trim remaining PCR primers. Read mapping against the human genome (hg19) was done using HISAT2 (v. 2.1.0, [24]) and fragment quantification was done using stringtie (v. 2.1.1, [25]). Differential gene expression analysis was performed using the DESeq2 R package [26]. The Wald test was performed for pair-wise comparison, and genes were considered significantly differentially expressed if absolute value of their log2 fold change was over 1 and if FDR was less than 0.05.

### 2.4. Proliferation Assay

OS cell lines, treated with increasing concentrations of PR619 for 14 h, 24 h and 48 h, were fixed with 1% glutaraldehyde and stained using crystal violet. Crystal violet staining was solubilized in Sorenson solution and absorbance was measured with a Victor² apparatus (Perkin Elmer, Villebon-sur-Yvette, France) at 570 nm.

### 2.5. Osteosarcoma Tissue Microarray

Tissue microarray (TMA) glass slides containing duplicate formalin fixed, paraffin embedded (FFPE) OS specimens (*n* = 40) were purchased from US Biomax, Inc. (Rockville, MD, USA). Characteristics of each sample of the TMA are listed in Table 2. TMA samples were stained for USP43, USP41, USP27x and USP6 antibodies (Life Technologies). After backing slides for 1 h at 60 °C, samples were processed routinely via deparaffinization, rehydration, antigen retrieval in 10 mM citrate buffer pH6 for 20 h at 60 °C. The endogenous peroxidase activity was then quenched, and non-specific binding was subsequently blocked with 2% (*v*/*v*) normal goat serum (Jackson ImmunoResearch) and 1% (p/v) B.S.A. in 1X TBS Tween pH 7.4 for 25 min at room temperature. Samples were then incubated with appropriate antibody diluted at 1:50 in blocking solution for 1 h at room temperature. Negative control was achieved by omitting a primary antibody. Following the primary antibody incubation, sections were washed in 1X TBS Tween pH 7,4 and samples were incubated for 45 min at room temperature using a polyclonal biotinylated goat anti-rabbit secondary antibody (Agilent), and finally with an HRP conjugated streptavidin complex (Agilent) for another 45 min at room temperature. Revelation was conducted with DAB liquid chromogen (microm-microtech France) and counterstained with Gill-, dehydrated through graded ethanol baths, cleared in OTTIX-plus and cover-slipped with Pertex mounting media.

Sections were analyzed using standard light microscopy. Each sample was examined twice by two different pathologists blinded to the experiment. The expression and distribution of USPs were assessed within the biopsies. Staining intensity was scored against the highest stained section of the slide sections. Grading of USPs expression was thus assessed as follows: no expression, low expression, intermediate expression and high expression. The expression ranking was done by giving a value for each expression level: 0 (no expression) to 3 (high expression). In the case of discordance between IHC staining grades on duplicate tissue sections, the highest grade was selected as the overall score for the tissue sample.

### 2.6. Annexin V Assay and Caspase Activity

OS cells were cultured and treated with or without PR619. Cell apoptosis was measured by flow cytometry (Cytomics FC500; Beckman Coulter, Roissy, France) according to the FITC Annexin V Apoptosis Detection Kit I protocol (BD Biosciences). Caspase3/7 activity was measured using the Apo-ONE caspase 3/7 kit from Promega (Charbonnière les bains, France).

### 2.7. Western Blot Analysis

Cells were lysed in lysis buffer (SDS 1%, Tris pH 7.4 10 mM, Sodium orthovanadate 1 mM). Samples in Laemmli buffer (62.5 mM Tris–HCl, pH 6.8, 2% SDS, 10% glycerol, 5% 2-mercaptoethanol, 0.001% bromophenol blue) containing equal amounts of total protein extracts were separated by SDS-polyacrylamide gel electrophoresis (SDS-PAGE). After transfer to PVDF membranes (Thermo Scientific, Waltham, MA, USA), membranes were probed with primary antibodies PARP, β-tubulin (Cell signaling, Leiden, The Netherland) or ubiquitin (Abcam, Paris, France). Membranes were then probed with secondary fluorescent antibodies (Li-COR IRDye). Antibody binding was visualized with the LI-COR odyssey Fc system (Cambridge, UK).

### 2.8. Osteosarcoma (OS) Mouse Model

Four-week-old female Rj:NMRI nude mice (Elevages Janvier, Le Genest Saint Isle, France) were used for in vivo experiments in accordance with the institutional guidelines of the Ethical Committee (CEEA Pays de la Loire no.06) and the authorization of the French Ministry of Agriculture and Fishery (Apafis # 8405-2017010409498904). Anesthetized mice received a paratibial injection of 1.5·10^6^ HOS cells, leading to a rapidly growing tumor in bone tissue. Once their tumors were palpable, mice were randomly assigned to control (vehicle) or PR619 groups. Tumor volume was measured twice a week. Mice were sacrificed when the tumor volume reached 2500 mm^3^ for ethical reasons and tumor fragments were collected. Under these conditions, pulmonary metastasis developed when tumor volumes were ≥2000 mm^3^. Lung metastases were analyzed and counted when the tumor volumes were equal to around 2500 mm^3^ in each mice group.

### 2.9. Cell Transfection with SiRNA against USPs

OS cells were transfected with Lipofectamine RNAiMAX (Invitrogen, Waltham, MA, USA) according to the manufacturer’s recommendations. SiRNA against USP6, USP27x and USP43 were supplied by Santacruz (Clinisciences, Nanterre, France). SiRNA against USP41 was supplied by Dharmacon (Horizon, Perkin Elmer).

### 2.10. Migration Assay

OS cells (60,000 cells/insert) were seeded onto the upper surface of transwell inserts (Falcon, Franklin Lakes, NJ, USA) and incubated at 37 °C for 8 h. At the end of the incubation period, cells on the upper surface of the inserts were wiped off, and the cells on the underside of the membrane were fixed, stained with crystal violet and counted by bright-field microscope.

### 2.11. Statistical Analysis

Statistical analyses were performed using GraphPad Prism 6 software (GraphPad Software, La Jolla, CA, USA). The Mann–Whitney test was used for in vitro and in vivo analyses. Results are given as means ± SD. Results with *p* < 0.05 were considered significant.

## 3. Results

### 3.1. Elevation of USPs Gene Expression in OS Cell Lines Compared to Mesenchymal Stem Cells

To assess the relevance for targeting USPs in OS, we first compared the gene expression of the 56 USPs described in the literature (USP1 to USP56) in seven OS cell lines (G292, MG63, HOS, KHOS, SAOS2, SJSA1 and U2OS) with that measured in MSCs differentiated (OB) or not into osteoblasts using RNA-sequencing analysis. As shown in Figure 1A,B, RNA-sequencing analysis shows that among USPs with altered expression in OS cells, four USPs (USP6, USP27x, USP41 and USP43) are significantly overexpressed in OS cells compared with MSCs differentiated or not into osteoblasts. To confirm these later results, we performed RT-qPCR. As shown in Figure 1C, the expression of these four USPs, USP6, USP27x, USP41 and USP43, is significantly increased in the majority of OS cell lines (HOS, MG63 and U2OS) compared to the expression measured in MSCs from three healthy donors.

### 3.2. Expression of USP6, USP27x, USP41 and USP43 in OS Patient Biopsies

IHC analysis demonstrate that all four USPs are expressed in the majority of OS samples (Figure 2A). Specifically, very few biopsies show no expression of USPs USA, 5 of 39 (12.8%), 6 of 39 (15.4%) and 3 of 39 (7.7%) for USP27x, USP41 and USP43, respectively. USP6 was detected in all biopsies. In addition, 13 of 39 (33.3%), 7 of 39 (18%), 7 of 39 (18%) and 21 of 39 (5.8%) biopsies have high expression of USP6, USP27x, USP41 and USP43, respectively. Further, 4 of 39 (10.2%), 12 of 39 (30.1%), 13 of 39 (33.3%) and 7 of 39 (17.9%) samples show weak expression of USP6, USP27x, USP41 and USP43, respectively. Remaining biopsies show intermediate expression of USP6, USP27x, USP41 and USP43. Most USPs are expressed in the cytoplasm of OS cells, 67%, 94%, 76% and 79% for USP6, USP27x, USP41 and USP43, respectively (Figure 2B). Interestingly, few OS cells also show nuclear expression of USPs, 32%, 6%, 14% and 21% for USP6, USP27x, USP41 and USP43, respectively. Of note, no significant correlation was observed between the expression or location of USP6, USP27x, USP41 and USP43, and the patient age, patient gender, or grade and histological subtypes of OS Appendix A.

### 3.3. Correlation between USP6 and USP41 Gene Expression and Patient Survival

To estimate the prognostic impact of USP6, USP27x, USP41 and USP43 expression, Kaplan Meier analysis of tumor samples from OS patients was performed using the R2 Genomics Analysis and Visualization Platform (http://r2.amc.nl, accessed on 26 August 2021).

Genome-wide gene expression analysis of high-grade OS are from GSE42352 (https://www.ncbi.nlm.nih.gov/geo/query/acc.cgi?acc=GSE42352, accessed on 26 August 2021).

As shown in Figure 3, the probability of overall survival is significantly reduced for patients with high expression of USP41 (Figure 3A) compared with those with low expression. In addition, the probability of metastasis-free survival is significantly reduced for patients with high expression of USP41 (Figure 3B) or USP6 (Figure 3D) compared with those with low expression. No significant correlation was observed between the probability of overall survival or the probability of metastase-free survival and USP27x expression. It should be noted that we do not have data for USP43.

### 3.4. PR619 Inhibits Primary Tumor Growth and Lung Metastases Development in an Orthotopic Model of OS

As it has been previously demonstrated that several USPs such as USP1, USP7, USP22 and USP39 participate in the control of proliferation and/or invasion of OS cells [19,20,21,22], and we show above that the expression of four other USPs, USP6, USP27, USP41 and USP43, is increased in OS cells, we then assessed the effects of the panUSPs inhibitor PR619 on primary OS tumor growth and lung metastases development. Of note, a pilot experiment using USP6, USP27x, USP41, or USP43 siRNAs suggests that USP6, USP27x, and USP43, are involved in the control of OS cell viability (Appendix A) and USP6, USP27x and USP41, in the ability of OS cells to migrate (Appendix A). The crucial role played by the bone microenvironment in OS tumor growth prompted us to use an orthotopic mouse model to evaluate the impact of PR619 on primary tumor growth and pulmonary metastasis development.

The efficacy of PR619 was first assessed by its ability to increase polyUbiquitinated (polyUb) proteins accumulation. As expected, treatment of OS cells (HOS, MG63 or U2OS) with PR619 (15 μM) increases the levels of polyUb proteins (Figure 4A). As shown in Figure 4B,C, treatment of mice with PR619 significantly reduces tumor growth in a dose-dependent manner. While the 0.5 mg/kg dose does not significantly affect tumor growth, the doses of 2 mg/kg and 10 mg/kg significantly inhibit the primary OS tumor growth (Figure 4B). Indeed, the mean tumor size at day 49 was 1705 ± 470 mm^3^ when the mice were treated with vehicle (control group) and only 1166 ± 435 mm^3^ and 1133 ± 522 mm^3^ when the mice were treated with 2 mg/kg and 10 mg/kg of PR619, respectively (Figure 4C). Interestingly, treatment of mice with 2 mg/kg or 10 mg/10/kg PR619 significantly repress lung metastases development. Indeed, the mean metastases number, when the tumor volumes were around 2500 mm^3^, was 14.7 ± 6.3 when the mice were treated with vehicle (control group) and only 7.5 ± 2.1 and 7.2 ± 3.1 when the mice were treated with 2 mg/kg and 10 mg/kg of PR619, respectively (Figure 4D). It can be noted that at these doses of PR619 (2 mg/kg and 10 mg/kg), PR619 does not affect the weight of mice.

### 3.5. PR619 Induces In Vitro Cell Death

To investigate the cellular mechanisms behind the effects of PR619 on OS tumor growth, we then performed in vitro experiments. Firstly, our results show that PR619 significantly decreases the viability of three OS cell lines, HOS, MG63 and U2OS, in a dose- and time-dependent manner (Figure 5A,B). The most sensitive cells are MG63 with an IC50 of 1.69 μM after 48 h of treatment and 2.89 μM and 3.70 μM for HOS and U2OS cells, respectively.

Next, we evaluated whether inhibition of OS cell viability by PR619 correlates with stimulation of cell apoptosis. Using annexin V/PI cytometric analysis, we could demonstrate that PR619 causes early and late apoptotic events in a dose-dependent manner (Figure 6A). For example, the percentage of HOS cells in early apoptosis (Annexin V+/PI-) is 2.9% in the absence of PR619, and reaches 35.9% and 26.2% after 24 h treatment of cells with 5 μM and 10 μM PR619, respectively. The percentage of HOS cells in late apoptosis (Annexin V+/PI+) is 8.3% in the absence of PR619, and reaches 16.1% and 51.9% after 24 h treatment of cells with 5 μM and 10 μM PR619, respectively. A difference in cell sensitivity can be observed, with MG63 cells being the most sensitive and U2OS cells the least sensitive. Indeed, the percentage of cells in early apoptosis is 48.8% for MG63 cells 48 h after treatment with 10 μM PR619 and only 26.2% and 16.8% for HOS and U2OS cells, respectively Fourthly, we measured the activity of apoptotic enzymes caspases-3 and -7 in OS cells treated 14 h with PR619 (5 μM or 10 μM). As shown in Figure 6B, PR619 significantly increases the activity of caspase 3/7. Caspase 3/7 activity is increased 2.18-fold, 2.55-fold and 1.63-fold after treatment of HOS, MG63 and U2OS cells with 5 μM of PR619 for 14 h and increased 4.46-fold, 6.35-fold and 2.53-fold after treatment of OS cells with 10 μM of PR619. Finally, we looked at the cleavage of PARP in OS cells treated with 5 μM or 15 μM PR619. Experiments show cleavage of PARP in all three OS cell lines after 24 h of treatment (Figure 6C).

## 4. Discussion

The poor clinical outcome of patients with OS underscores the need to develop new therapeutic strategies for this disease. In this context, the role of USPs in cancer is being increasingly studied. Recent works highlight that many USPs could be prime targets in cancer treatment due to their ability to regulate many key processes in tumor development such as primary tumor growth and the metastasis process [13,14,18,29,30,31].

In the present study, by comparing the expression of the 56 USPs described in the literature, in seven OS cell lines and in mesenchymal stem cells or osteoblasts, we identified four USPs: USP43, USP41, USP27x and USP6, with increased expression in OS tumor cell lines.

Tumor development can be schematically divided into two major stages: primary tumor growth and metastatic development. In this context, several USPs have been identified as participating in these two stages of OS tumor development.

Regarding the growth of OS primary tumor, USP1, USP22 and USP39 have been shown to participate in the control of OS cell proliferation [19,21,22]. Among the four USPs that we have shown increased expression in OS cells, USP6 is described as able to modulate cell proliferation. Historically, the gene encoding this enzyme was initially cloned in primary bone tumors: Ewing’s sarcoma [32]. Most works indicate a close correlation between USP6 activity and activation of the NFkB signaling pathway [32,33,34]. In this context, it has been shown that NFκB subunits (RelA, RelB, and c-Rel) are activated and abundant in the nucleus of OS cells such as U2OS, MG63, Saos-2, and HOS cells compared to hFOB1.19 and that activation of the NFkB pathway is associated with increased proliferation of OS cells [35]. USP27x and USP41 have also been described to modulate cell proliferation in cancer cells. For example, USP27 through stabilization of cyclin E participates in the control of primary liver tumor growth [36], and USP41 regulates the proliferation of lung cancer cells [37]. USP43 also seems to participate in the control of colorectal cancer cell proliferation [38]. Thus, it can be hypothesized that USP6, USP27x, USP41 and USP43 via various mechanisms would be able to modulate OS cell proliferation and thus primary tumor growth.

Regarding the metastatic development of OS, USP1 and USP22 participate in the control of migration, invasion of OS cells and thus metastatic development [19,21]. One of the mechanisms of action identified to explain these processes is the ability of various USPs to modulate the epithelial-mesenchymal transition (EMT). For example, USP7 regulates the metastatic development of OS via modulation of EMT [20]. Regarding the four USPs that we have shown increased expression in OS cells, USP27x and USP43 are involved in the control of EMT and thus metastatic progression in various cancers. For example, high expression of USP43 in breast and colon cancers has been shown to be associated with tumor development by increasing the ability of tumor cells to migrate. These effects are closely correlated with the ability of these proteins to stimulate EMT [38,39]. These data suggest that in OS in which we show increased expression of USP43, this protein may participate in the regulation of EMT, a cellular process largely involved in OS progression. Indeed, despite OS arise from transformed cells of mesenchymal origin, numerous studies demonstrated that an overexpression of EMT-transcription factors such as Snails, ZEBs or Twist allow an EMT-like phenomena that promotes the invasive properties of OS cells [40,41]. Interestingly, ZEB-1 whose expression is strongly increased in OS tissues compared to normal bone tissues, and in patients with lung metastases [42] is deubiquitinated and thus stabilized by USP43 [38]. Regarding USP27, it has been recently shown that this USP is able to regulate EMT by promoting the stabilization of Snail [43]. Interestingly, this transcriptional factor is expressed in the three major histological subtypes of long bone osteosarcoma (osteoblastic, chondroblastic and fibroblastic) [44]. Of note, our work using OS TMA indicates that USP27x is expressed in all histological subtypes of OS. In addition, USP41 also regulates the ability of lung cancer cells to migrate [37].

Although there are more or less specific inhibitors for some USPs such as USP1, USP2, USP4, USP7, USP8, USP9X, USP10, USP14, USP20 or USP30 [45], there is no specific inhibitor for USP43, USP41, USP27x or USP6 that is currently available. In this context, to validate the relevance to target USPs in OS, we used the pan-USPs inhibitor PR619 in vivo on the growth of OS tumors using an orthotopic mouse model. Our results demonstrate that this inhibitor affects the growth of the primary tumor by acting primarily on cell death. Signaling pathways able to induce apoptosis have been schematically classified into: (1) extrinsic pathways initiated by death receptors and (2) intrinsic pathways initiated by mitochondrial events. In this work, we demonstrated that PR619 induce the cleavage and activation of caspase-3, a common mediator of both apoptotic pathways. In addition, we demonstrated that PR619 induce the cleavage of PARP, a substrate of caspase-3 known as an indicator of DNA damage and apoptosis. Whatever the exact involvement of the intrinsic and/or extrinsic pathways, we clearly demonstrated that PR619 induces in vitro OS cell apoptosis and thus inhibits the in vivo tumor growth. In this context, many USPs have been shown to participate in the control of cell death through regulation of expression or degradation of the tumor suppressor p53 [14].

We also showed that PR619 reduced metastatic development. It can be hypothesized that PR619 acts via the inhibition of USPs whose expression is increased in OS cells and involved in these processes such as USP1 and USP22 [19,21] or potentially involved in these processes such as USP6, USP27x and USP41. It can also be hypothesized that PR619 acts via the inhibition of USPs whose expression is not necessarily increased in OS cells, but which are able to modulate the activity of signaling pathways involved in the regulation of EMT. For example, the TGF-β signaling pathway is regulated by USP4 and USP15. In particular, these enzymes stabilize the TBRI receptor and thus increase the ability of TGF-β to stimulate cell migration and therefore metastatic development in liver cancers and in glioblastoma [46,47,48]. This seems particularly important in the context of OS, where the major role of TGF-β in the development of pulmonary metastasis has been shown [49,50,51].

Although the use of PR619 seems impossible in the clinic due to the targeting of all USPs and therefore potentially carrying very important side effects, our work demonstrates the rationale of targeting USPs in OS.

## 5. Conclusions

Our work shows for the first time the expression of four new USPs in OS, USP43, USP41, USP27x and USP6. In particular, the expression of two of them, USP41 and USP6, is correlated with patient survival. The use of an inhibitor, PR619, indicates the importance of targeting this family of proteins in OS.

## Figures and Tables

**Figure 1 cells-10-02268-f001:**
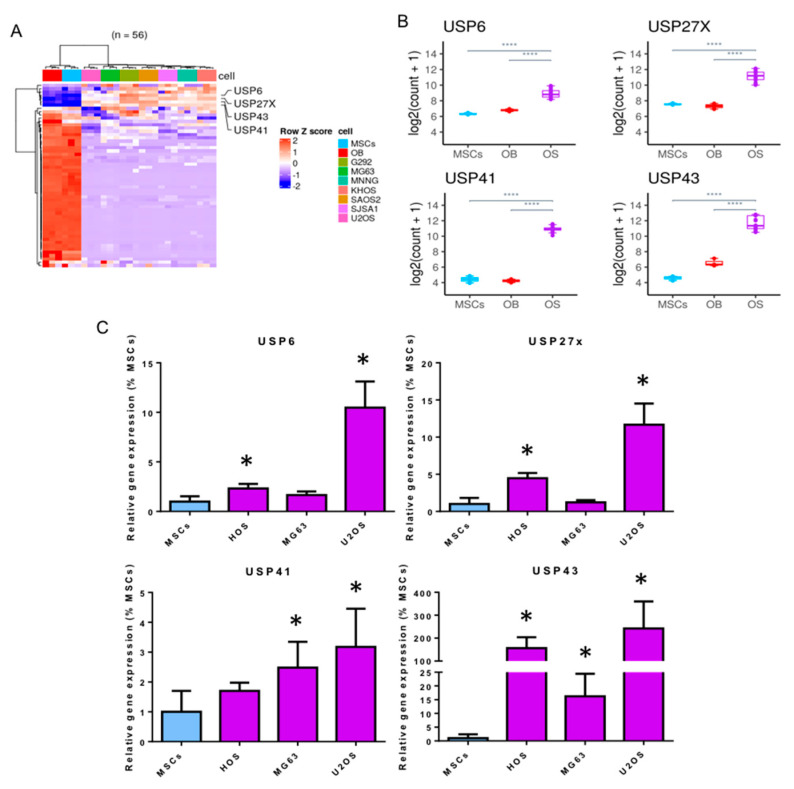
Increased expression of USP6, USP27x, USP41 and USP43 in OS cell lines. (**A**) Heatmap of the 56 USP genes in MSCs, osteoblasts (MSCs treated with differentiation medium up to 14 days after proliferation phase) and OS cell lines (KHOS, 143B, HOS, MG63, G292, U2OS, SAOS2). Counts were variance stabilized and transformed to z-scores. (**B**) Normalized expression levels of USP genes (USP6, USP27x, USP41, USP43) in MSCs, osteoblasts (MSCs treated with differentiation medium up to 14 days after proliferation phase) and OS cell lines (KHOS, 143B, HOS, MG63, G292, U2OS, SAOS2). OS cell lines were gathered together within the OS group and pair-wise gene expression comparison was performed using Wald Test. All of the genes represented here are significantly differentially expressed (|Log2 FC| > 1, **** adj. *p* value ≤ 0.0001) in comparison OS vs. MSCs and OS vs. OB (MSCs differentiated into osteoblasts). (**C**) MSCs from healthy volunteers and three OS cell lines (HOS, MG63 and U2OS) were cultured. At 80% confluence, RNAs were extracted and USP6, USP27x, USP41 and USP43 mRNA steady-state levels were quantified by RT-qPCR analysis (bars indicate means ± SD of 3 independent experiments, each performed in triplicate, * *p* < 0.05).

**Figure 2 cells-10-02268-f002:**
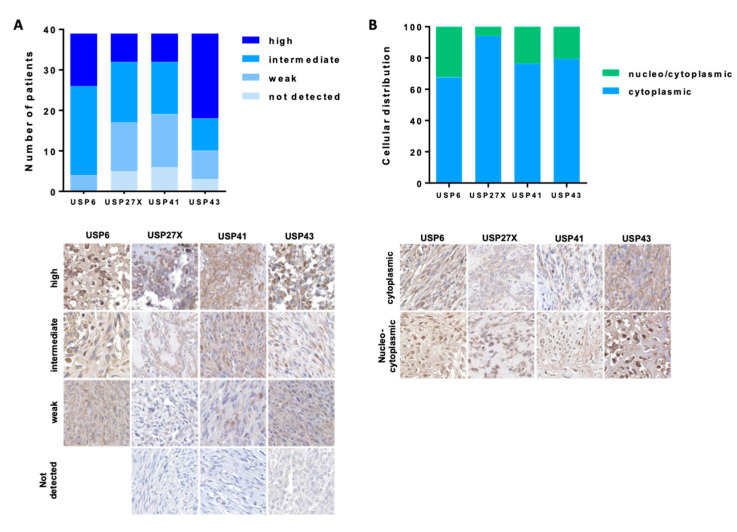
*Expression of USP43, USP41, USP27x* and *USP6* in *OS TMA* OS. Tissue microarray (TMA) glass slides containing 40 OS specimens were stained with anti USP6, USP27x, USP41 and USP43 antibodies. Samples were analyzed using standard light microscopy by two different pathologists blinded to the experiment. Expression (**A**) and distribution (**B**) of USPs was assessed within the biopsies as described in the Materials and Methods section. USPs expression was assessed as follows: no detected expression, weak expression, intermediate expression and high expression.

**Figure 3 cells-10-02268-f003:**
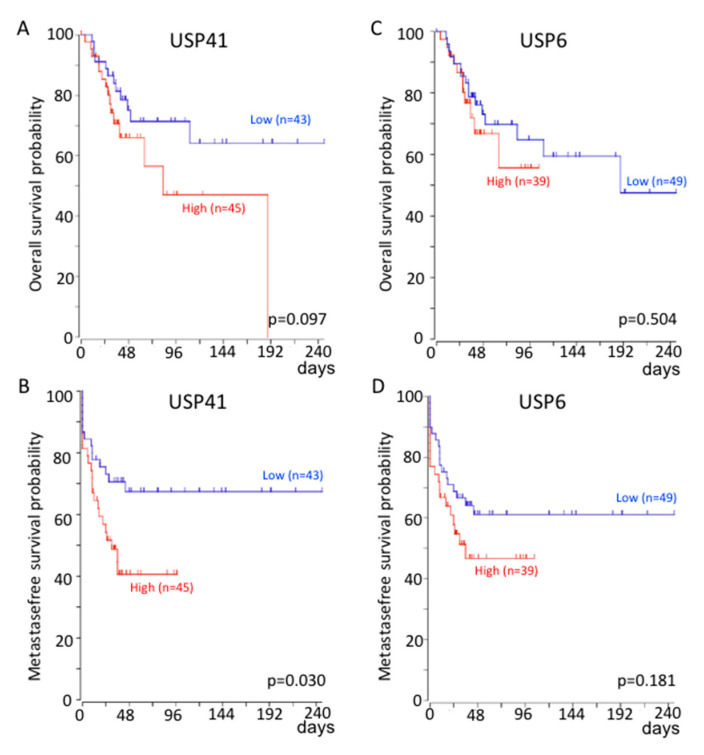
Correlation between USP41 and USP6 expression with patient survival probability. Kaplan–Meier analysis of the survival outcome of patients dichotomized into high and low relative to average expression level of USP41 (**A**,**B**) or USP6 (**C**,**D**) levels, following analysis of expression profiles GSE42352 [27,28] from an OS patient cohort comprising 88 samples. Analysis was performed using R2 (http://r2.amc.nl, accessed on 26 August 2021). The *p* value is from log-rank tests.

**Figure 4 cells-10-02268-f004:**
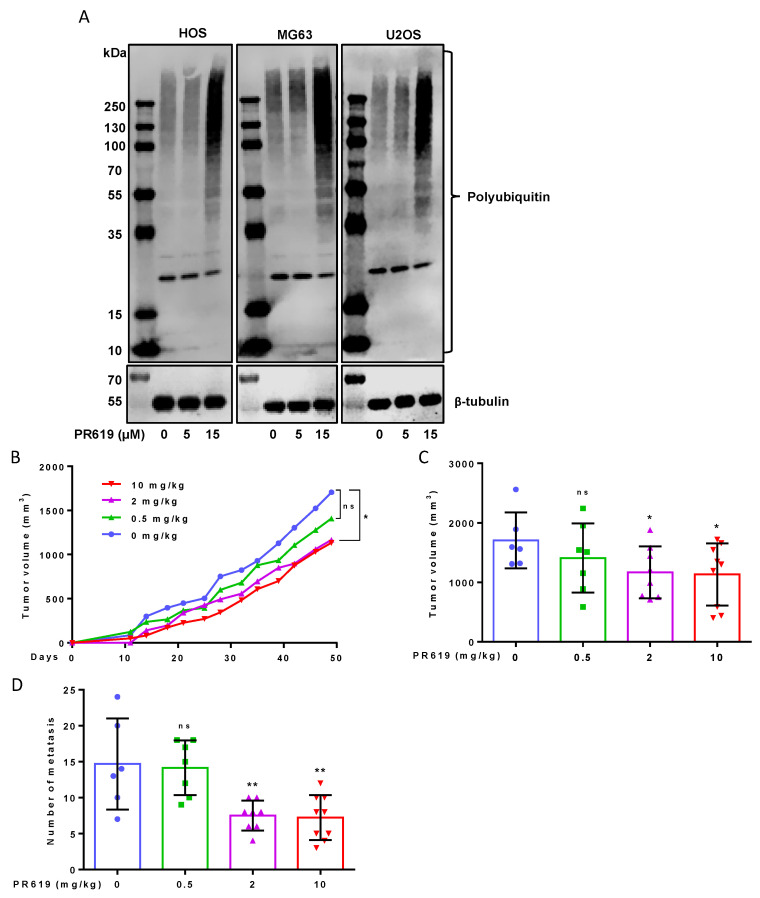
*PR619 reduces in vivo primary tumor growth and lung metastases development*. (**A**) OS cells were treated or not with 5 or 15 μM PR619, as indicated, for 24 h. After incubation, the levels of polyUb proteins were measured by Western blot analysis as described in Materials and Methods section. Representative blots of three independent experiments are shown. (**B**–**D**) Intramuscular paratibial injections of 1.5 × 10^6^ HOS cells, leading to a rapidly growing tumor in bone tissue, were performed in three groups of nude mice treated with vehicle (blue) or PR619 (Green: 0.5 mg/kg, Purple: 2 mg/kg, Red: 10 mg/kg). (**B**) Tumor volumes were measured two times per week for seven weeks. (**C**) Bars indicate means tumor volumes of each group measured 49 days after cell injection (mean ± SD; * *p* < 0.05). (**D**) The number of lung metastases were counted when the tumor volumes were approximately 2500 mm^3^ (mean ± SD; ** *p* < 0.01).

**Figure 5 cells-10-02268-f005:**
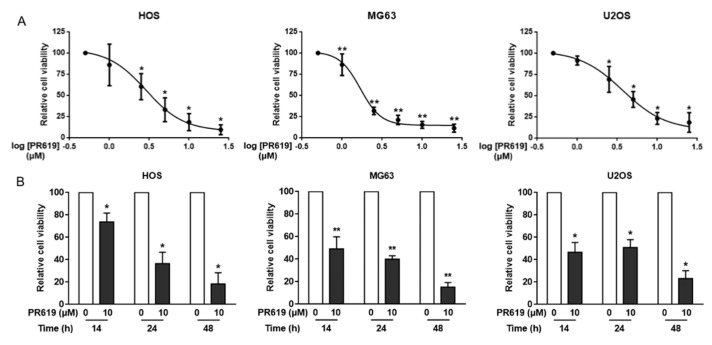
*PR619 reduces* in vitro *cell proliferation*. (**A**) Three OS cell lines (HOS, MG63 and U2OS) were treated or not for 48 h with PR619 (1, 2.5, 5, 10 or 25 μM). Cell viability was evaluated as described in the Materials and Methods section using crystal violet staining. For each cell line, the graph indicates the relative cell number compared with untreated cells. The mean ± SD of at least four independent experiments, each performed in sextuplicate, is presented (* *p* < 0.05, ** *p* < 0.01). (**B**) Three OS cell lines (HOS, MG63 and U2OS) were treated or not for 14 h, 24 h or 48 h with 10 μM PR619. Cell viability was evaluated as described in the Materials and Methods section using crystal violet staining. For each cell line, the graph indicates the relative cell number compared to untreated cells. The mean ± SD of at least four independent experiments, each performed in sextuplicate, is presented (* *p* < 0.05, ** *p* < 0.01).

**Figure 6 cells-10-02268-f006:**
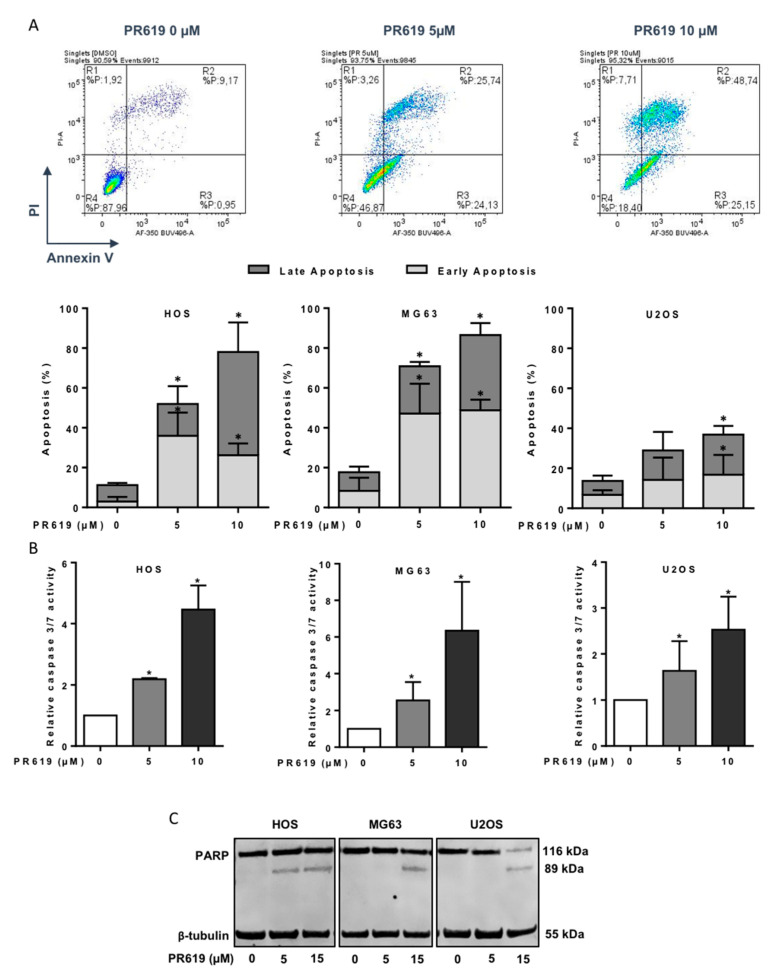
*PR619 induces* in vitro *cell death in OS cell lines*. (**A**) Upper panels: Representative dot plots of HOS cells treated or not with 5 or 10 μM PR619 for 24 h are shown (representative graphs of three independent experiments). Lower panels: OS cells were treated or not with 5 or 10 μM PR619 for 24 h. Bars indicate the means ± SD of the relative number of cells in early- or late-phase apoptosis (3 independent experiments) (* *p* < 0.05). (**B**) OS cells were treated with 5 or 10 μM PR619 for 14 h. Relative caspase-3/7 activity was measured as described in the Materials and Methods section. Bars indicate the caspase-3/7 activity (mean ± SD) of three independent experiments, each performed in triplicate (* *p* < 0.05). (**C**) OS cells were treated or not treated with 5 or 15 μM PR619, as indicated, for 24 h. After incubation, PARP cleavage levels were detected by Western blot analysis as described in the Materials and Methods section. Representative blots of three experiments are shown.

**Table 1 cells-10-02268-t001:** Primers sequences.

	Forward Primers	Reverse Primers
B2M	TTCTGGCCTGGAGGCTATC	TCAGGAAATTTGACTTTCCATTC
HPRT	TGACCTTGATTTATTTTGCATACC	CGAGCAAGACGTTCAGTCCT
USP6	CATGCCATCTCTTCCTGACAGC	CAATGGCATTCCAAAGAGGCTGG
USP27X	ACCAAGGAACCTTGGAGAGTGG	CCTTCACTGTCCAGTACGTCCT
USP41	TGAATGTGGACTTCGCCAGG	ATGTTGGACAAACAGGGGCA
USP43	TGGGCATTACACAGCCTACTG	AGACAGGGAGGAGCTGGTAG

**Table 2 cells-10-02268-t002:** Clinical data on samples.

Patient	Core Position	Age	Gender	Localisation	Stage	Fibroblastic Osteosarcoma	Osteo Chondroblastic Osteosarcoma	Mixte Osteosarcoma
1	A1/A2	13	F	femur	IB		+/+	
2	A3/A4	12	F	femur	IIB	+/+		
3	A5/A6	23	F	femur	IIB			
4	A7/A8	21	F	femur	IB	+/+		
5	A9/A10	42	F	femur	IIB	+/+		
6	B1/B2	16	F	femur	IIB	+/+		
7	B3/B4	15	F	femur	IB	+/+		
8	B5/B6	35	F	femur	IB		+/+	
9	B7/B8	64	F	femur	IB		+/+	
10	B9/B10	29	F	femur	IA		+/+	
11	C1/C2	30	F	femur	IB		+/+	
12	C3/C4	51	F	femur	IB		+/+	
13	C5/C6	11	F	femur	IA		+/+	
14	C7/C8	47	F	femur	IB		+/+	
15	C9/C10	47	F	femur	IIB			+/+
16	D1/D2	16	F	femur	IB		+/+	
17	D3/D4	12	F	femur	IIB		+/+	
18	D5/D6	16	F	femur	IB			
19	D7/D8	16	F	femur	IIB		+/+	
20	D9/D10	37	F	femur	IA		+/+	
21	E1/E2	14	F	femur	IIB		+/+	
22	E3/E4	14	F	femur	IIB	+	/+	
23	E5/E6	32	F	femur	IIB	+/+		
24	E7/E8	46	F	femur	IB		+/+	
25	E9/E10	16	F	femur	IIB		+/+	
26	F1/F2	17	F	tibia	IB		+/+	
27	F3/F4	17	F	tibia	IB		+/+	
28	F5/F6	14	F	tibia	IIB		+/+	
29	F7/F8	38	F	tibia	IIB	+/+		
30	F9/F10	20	F	tibia	IIB		+/+	
31	G1/G2	60	F	tibia	IIB		+/+	
32	G3/G4	31	F	tibia	IIB		+/+	
33	G5/G6	18	F	tibia	IB		+/+	
34	G7/G8	32	F	tibia	IB			+/+
35	G9/G10	41	F	tibia	IA		+/+	
36	H1/H2	30	M	rib	IA	+	/+	
37	H3/H4	32	M	rib rib	IIA	+/+		
38	H5/H6	33	M	upper jaw	IA			+/+
39	H7/H8	15	M	upper jaw	IVB	+		/+
40	H9/H10	51	M	fibula	IIB		+/+	
	H11	42	M	Pheochromocytoma				

## Data Availability

The data presented in this study are available on request from the corresponding author or from Benjamin Ory (benjamin.ory@univ-nantes.fr) for RNAseq data.

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
