# Peer review of "Overexpression of the Ubiquitin Specific Proteases USP43, USP41, USP27x and USP6 in Osteosarcoma Cell Lines: Inhibition of Osteosarcoma Tumor Growth and Lung Metastasis Development by the USP Antagonist PR619"

_cells, 2021, doi:10.3390/cells10092268_

Round 1

Reviewer 1 Report

Lavaud et al

Ubiquitin-Specific Proteases as novel anticancer targets in osteosarcoma?

The authors show that USP expression is induced in osteosarcoma cells and that the activity of the USPs protects from cell death. They have compared USP expression in both osteosarcoma cell lines and mesenchymal stem cells, and in patient biopsies. They relate USP expression to patient survival and can reduce tumour size in an orthotropic mouse model. Finally, they show that a panUSP inhibitor can induce apoptosis in cultured osteosarcoma cells. While the data presented is novel, important and convincing, no mechanistic molecular explanation for the protective function of USPs in osteosarcoma is provided.

I find the experimental work to be sufficient to prove the hypothesis stated. However, I find the manuscript language and style not to be sufficient for publication. The abstract does not form a coherent entity sufficiently presenting the research. The current discussion consists of separated statements on USPs roles in cancers and needs to be rewritten to be reasonable and conclusive. Also figure legends could be improved by including more details and result descriptions would benefit from more explanatory. Finally, phrasing the title as a question is not very convincing.

Reviewer 2 Report

  1. The title need to amend to be more specific with key findings of current work. The title provided here is more like a review article.
  2. Discussion and Conclusion sections need to further improved, e.g. would be advisable to draw up a figure to clearly illustrate and summarize the whole (or part of) conclusion in a ‘pictorial’ manner.
  3. In Figure 2, authors showed protein expression of USP43, USP41, USP27x and USP6. How about mRNA levels of these targets in patients?
  4. Authors need to give proper rational why in Fig3 they only target USP41 and USP6 instead of all four USPs (USP43, USP41, USP27x and USP6).
  5. Authors claimed that four USPs identified here. But to verify the corresponding biological functions in OS patients, they should at least choose two targets for specific RNAi knockdown. PR619 treatment is simply not sufficient for clarification of underlying mechanisms involved in.

Reviewer 3 Report

The manuscript presents interesting observations concerning expression of several USPes in osteosarcoma. The manuscript is well written with a good review of data related to expression of these proteases in a context of cancer.

I would suggest several improvements that are necessary to make with work complete.

  1. There is no explanation how many USPes were tested prior to the final list of four. They were more expressed than other USPes but how much? Which were tested? At this moment it seems a very subjective choice.
  2. Expression data should be presented as an expression array for each cell line. Pooling them together limits their use.
  3. Figure 2 does not explain criteria used for the expression assessment in tissue samples.
  4. Data in Figure 5A should be presented as a scatter plot in a log concentration scale.
  5. There is a very modest effect on primary tumor size after PR619 treatment which is not commented.
  6. Indeed, PR619 is a "pan" inhibitor for USPes. It would be beneficial to test other inhibitors of USPes to make data more convincing.
  7. It would be good to have a prove that these proteases are indeed affected directly by PR619 (it is a reversible inhibitor).
  8. polyUB accumulation resulting from USPes inhibition should be included.
  9. patient data are not really used except in a form of hard to read list. There is sufficient number of patients included in this study to attempt some initial bioinformatics analysis.
  10. The main question would be: are these particular USPes work as a team or this is just a random set (hard to believe in the latter)?
  11. status of proteasome (at least its activity) should be also included.
  12. Discussion calls for a better general context of USPes in cancer: what do they do that they influence cancer development?

Reviewer 4 Report

In the current manuscript, the authors have explored the correlation of deubiquitinases' (DUBs) expression with osteosarcoma and went to provide a proof of concept for the potential use of inhibitors targeting DUBs to treat osteosarcoma. 

The conclusions drawn by the authors are mostly supported by the findings presented in the manuscript. However, I have few minor comments. 

Overall I recommend the manuscript for minor revisions.

Comments:

 1. The authors have not shown any ubiquitin blots of the PR619 treated cells to show that the inhibitor works. I think it is required.

2. Fig 6: For both early and late apoptosis the text mentions AnnexinV+/PI-. Please correct that. The statistical analysis for the bar graphs are not provided.

Round 2

Reviewer 2 Report

Thank for the amended version of ms. 

Author Response

We thank the reviewer for the positive feedback on our work “Thank for the amended version of ms”

In terms of the presentation of the results, we have better commented the results presented in Figures 5 and 6.

Figure 5: We have calculated the IC50s for each cell, HOS, MG63 and U2OS. A sentence has been added " The most sensitive cells are MG63 with an IC50 of 1.69 mM after 48h of cell treatment and 2.89 mM and 3.70 mM for HOS and U2OS cells, respectively ».

Figure 6 : We have detailed the results concerning early and late apoptosis as well as the activity of capspases 3 and 7. Two paragraphs have been added

« A difference in cell sensitivity can be observed, with MG63 cells being the most sensitive and U2OS cells the least sensitive. Indeed, the percentage of cells in early apoptosis is 48.8% for MG63 cells 48h after treatment of the cells with 10 mM PR619 and only 26.2% and 16.8% for HOS and U2OS cells, respectively »

« Caspase 3/7 activity is increased 2,18-fold, 2,55-fold and 1,63-fold after treatment of HOS, MG63 and U2OS cells with 5 mM of PR619 for 14h. and increased 4.46-fold, 6.35-fold and 2.53-fold after treatment of OS cells with 10 mM of PR619.

We hope that all the changes made to our original manuscript satisfactorily address the reviewer’s issues and that this revised version is now acceptable for publication.

Reviewer 3 Report

The Authors responded to the presented to some concerns adequately. 

I would still look how the results are utilized: many of them are left without much of a comment only as just dry observations, for example fig. 6 results. In figure 5a data were transformed into a log scale but there are no IC50s calculated (apparently cell lines differ in their response to PR619). Figure 4A shows polyUb proteins but the clamp lumps all the proteins on the gel (way too much!). Finally, my understanding is that you are discussing polyUb proteins but you are using expression ubiquitination. This could be correct in certain context only. Please make sure and edit the text to address this concern.

Discussion has been improved but it is still somewhat detached from the findings.

Author Response

We thank the reviewer for the positive feedback on our work 

  • Concerning the exploitation of the results, we have further detailed the results presented in figures 5 and 6.

Figure 5: We have calculated the IC50s for each cell, HOS, MG63 and U2OS. A sentence has been added " The most sensitive cells are MG63 with an IC50 of 1.69 µM after 48h of cell treatment and 2.89 µM and 3.70 µM for HOS and U2OS cells, respectively ».

Figure 6 : We have detailed the results concerning early and late apoptosis as well as the activity of capspases 3 and 7. Two paragraphs have been added

« A difference in cell sensitivity can be observed, with MG63 cells being the most sensitive and U2OS cells the least sensitive. Indeed, the percentage of cells in early apoptosis is 48.8% for MG63 cells 48h after treatment of the cells with 10 µM PR619 and only 26.2% and 16.8% for HOS and U2OS cells, respectively »

« Caspase 3/7 activity is increased 2,18-fold, 2,55-fold and 1,63-fold after treatment of HOS, MG63 and U2OS cells with 5 µM of PR619 for 14h. and increased 4.46-fold, 6.35-fold and 2.53-fold after treatment of OS cells with 10 µM of PR619.

  • Regarding the ubiquitination of the proteins shown in figure 4A, we have corrected the text and now refer to the accumulation of polyUb proteins.
  • Regarding the discussion we have added a paragraph on PR619-induced cell death.

« Signaling pathways able to induce apoptosis have been schematically classified into (1) extrinsic pathways initiated by death receptors and (2) intrinsic pathways initiated by mitochondrial events. In this work, we demonstrated that PR619 induce the cleavage and activation of caspase-3 a common mediator of both apoptotic pathways. In addition, we demonstrated that PR619 induce the cleavage of PARP, a substrate of caspase-3 known as an indicator of DNA damage and apoptosis. Whatever the exact involvement of the intrinsic and/or extrinsic pathways, we clearly demonstrated that PR619 induces in vitro OS cell apoptosis and thus inhibits the in vivo tumor growth ».

We hope that all the changes made to our original manuscript satisfactorily address the reviewer’s issues and that this revised version is now acceptable for publication.